# Differences in Atrial Remodeling in Hypertrophic Cardiomyopathy Compared to Hypertensive Heart Disease and Athletes’ Hearts

**DOI:** 10.3390/jcm11051316

**Published:** 2022-02-27

**Authors:** Helge Servatius, Simon Raab, Babken Asatryan, Andreas Haeberlin, Mattia Branca, Stefano de Marchi, Nicolas Brugger, Nikolas Nozica, Eleni Goulouti, Elena Elchinova, Anna Lam, Jens Seiler, Fabian Noti, Antonio Madaffari, Hildegard Tanner, Samuel H. Baldinger, Tobias Reichlin, Matthias Wilhelm, Laurent Roten

**Affiliations:** 1Department of Cardiology, Inselspital, Bern University Hospital, University of Bern, 3010 Bern, Switzerland; simon.raab@gmx.ch (S.R.); babken.asatryan@insel.ch (B.A.); andreas.haeberlin@insel.ch (A.H.); stefano.demarchi@insel.ch (S.d.M.); nicolas.brugger@insel.ch (N.B.); nikolas.nozica@insel.ch (N.N.); eleni.goulouti@insel.ch (E.G.); elenageorgieva.elchinova@insel.ch (E.E.); annalam@gmx.ch (A.L.); jens.seiler@insel.ch (J.S.); fabian.noti@insel.ch (F.N.); antonio.madaffari@insel.ch (A.M.); hildegard.tanner@insel.ch (H.T.); samuel.baldinger@insel.ch (S.H.B.); tobias.reichlin@insel.ch (T.R.); matthias.wilhelm@insel.ch (M.W.); laurent.roten@insel.ch (L.R.); 2CTU Bern, University of Bern, 3010 Bern, Switzerland; mattia.branca@ctu.unibe.ch

**Keywords:** athlete’s heart, left ventricular hypertrophy, atrial cardiomyopathy, signal-averaged ECG, P-wave duration, LAVI

## Abstract

Background: Hypertrophic cardiomyopathy (HCM), hypertensive heart disease (HHD) and athletes’ heart share an increased prevalence of atrial fibrillation. Atrial cardiomyopathy in these patients may have different characteristics and help to distinguish these conditions. Methods: In this single-center study, we prospectively collected and analyzed electrocardiographic (12-lead ECG, signal-averaged ECG (SAECG), 24 h Holter ECG) and echocardiographic data in patients with HCM and HHD and in endurance athletes. Patients with atrial fibrillation were excluded. Results: We compared data of 27 patients with HCM (70% males, mean age 50 ± 14 years), 324 patients with HHD (52% males, mean age 75 ± 5.5 years), and 215 endurance athletes (72% males, mean age 42 ± 7.5 years). HCM patients had significantly longer filtered P-wave duration (153 ± 26 ms) and PR interval (191 ± 48 ms) compared to HHD patients (144 ± 16 ms, *p* = 0.012 and 178 ± 31, *p* = 0.034, respectively) and athletes (134 ± 14 ms, *p* = 0.001 and 165 ± 26 ms, both *p* < 0.001, respectively). HCM patients had a mean of 4.9 ± 16 premature atrial complexes per hour. Premature atrial complexes per hour were significantly more frequent in HHD patients (27 ± 86, *p* < 0.001), but not in athletes (2.7 ± 23, *p* = 0.639). Left atrial volume index (LAVI) was 43 ± 14 mL/m^2^ in HCM patients and significantly larger than age- and sex-corrected LAVI in HHD patients 30 ± 10 mL/m^2^; *p* < 0.001) and athletes (31 ± 9.5 mL/m^2^; *p* < 0.001). A borderline interventricular septum thickness ≥13 mm and ≤15 mm was found in 114 (35%) HHD patients, 12 (6%) athletes and 3 (11%) HCM patients. Conclusions: Structural and electrical atrial remodeling is more advanced in HCM patients compared to HHD patients and athletes.

## 1. Introduction 

Left ventricular hypertrophy is a proportional reaction of heart muscle cells to increased loading conditions, in patients with long-standing arterial hypertension or in long-term endurance athletes, leading to hypertensive heart disease (HHD) and athlete’s heart, respectively. Left ventricular hypertrophy disproportional to loading conditions occurs in patients with hypertrophic cardiomyopathy (HCM). Pathogenic variants in sarcomeric genes underlie nearly 65% of HCM cases, and in the vast majority are transmitted as an autosomal dominant trait [1,2,3,4,5,6,7]. Hypertrophic cardiomyopathy is diagnosed in the presence of left ventricular wall thickness ≥15 mm detected by any imaging technique [1,5,8]. In addition to structural features, electrocardiographic abnormalities are identified in 90% of HCM patients [9,10,11]. However, distinguishing HCM from other forms of left ventricular hypertrophy can be difficult because of the lack of specific electrocardiographic and echocardiographic diagnostic parameters, particularly when the interventricular wall thickness (IVS) is >12 mm in athletes or in patients with HHD [1,5,9,12].

Left ventricular hypertrophy increases left atrial wall stress by several mechanisms, thereby promoting atrial cardiomyopathy and ultimately atrial fibrillation. HHD, HCM and athlete’s heart have all been associated with an increased prevalence of atrial fibrillation [13,14]. Structural alterations in patients with HCM affect both the ventricular myocardium and the atrial myocardium, and hemodynamic strain imposed on the atria by the abnormal ventricular loading conditions further promotes atrial remodeling in patients with HCM [15,16,17]. In athletes, cumulative training hours and repetitive episodes of high atrial volume load increase atrial wall stress and atrial remodeling [18].

The aim of this study was to compare electrocardiographic (12-lead ECG, signal-averaged ECG (SAECG), 24 h Holter ECG) and echocardiographic parameters of atrial and LV function in patients with HCM, HHD and athletes.

## 2. Methods 

In this prospective observational investigator-initiated study, three different patient populations were prospectively enrolled (see details below). The Cantonal Ethics Committee of Bern, Switzerland, approved the study protocol and all participants provided written informed consent. The following exclusion criteria were applied: atrial fibrillation or atrial flutter, manifest ventricular pre-excitation, severe pulmonary hypertension, severe mitral valve disease, mechanical valve replacements, acute coronary syndrome, major cardiac surgery or hospitalization due to heart failure within the previous 3 months, atrial stimulation by either a pacemaker (PM) or an implantable cardioverter defibrillator (ICD), heart transplantation and incapacity to provide written informed consent.

### 2.1. Hypertrophic Cardiomyopathy Patient Population

Patients with HCM were enrolled in the outpatient HCM clinic. The diagnosis of HCM was based on internationally accepted criteria [1].

### 2.2. Hypertensive Heart Disease Patient Population

As a first comparison group, we used subjects with arterial hypertension and left ventricular hypertrophy included in the prospective STAR-FIB cohort study. Left ventricular hypertrophy was defined as a left ventricular myocardial mass index above normal range according to the published criteria (≥102 g/m^2^ in males and ≥88 g/m^2^ in females) [19]. The STAR-FIB cohort study was an atrial fibrillation screening study that included 795 patients without atrial fibrillation, aged 65–84 years [20,21].

### 2.3. Endurance Athlete Population

As a second comparison group, we used non-elite marathon runners included in two prospective, cross-sectional observational studies [22,23]. We performed the same, prospective examinations in all athletes as in patients with HCM, except for ambulatory 24 h Holter ECG, which we only performed in a subset of athletes, and laboratory examinations, which were not performed in athletes. We excluded athletes with hypertension (blood pressure ≥140/90 mmHg), a history of cardiovascular diseases, atrial fibrillation or atrial flutter.

### 2.4. Patient Work-Up

All patients from all 3 groups underwent the same baseline examination, including a comprehensive medical history, determination of heart rate and blood pressure at rest, 12-lead ECG, SAECG of the P wave and of the QRS complex, 24 h Holter monitoring, transthoracic echocardiography and laboratory analysis including high-sensitivity C-reactive protein (hsCRP), brain natriuretic peptide (BNP) and high-sensitivity Troponin T (hsTnT). ECGs and SAECGs were recorded with patients in a supine position at a sweep speed of 25 mm/s (MAC5500, GE Healthcare, Chicago, IL, USA). Methods for recording and analyzing the signal-averaged P wave have been described previously [24,25]. In brief, we recorded three bipolar orthogonal leads, referred to as the x, y and z leads, in a room without electrical interferences. Measurements computed by the system included filtered P-wave duration, root-mean-square voltage of the terminal 40, 30 and 20 milliseconds, root-mean-square voltage of the P wave and the P-wave integral (area under the vector magnitude curve from P-wave onset to offset). We performed standard transthoracic echocardiography (Vivid E95, GE Healthcare, Chicago, IL, USA) according to the recommendations of the European Association of Echocardiography [26]. The left ventricular ejection fraction (LVEF), dimensions and wall thickness as well as left atrial diameter and mono- and biplane-volume were determined in the parasternal long-axis view, 4- and 2-chamber view, as appropriate. The mitral valve inflow pattern was determined by pulsed-wave Doppler (E/A ratio; deceleration time). Myocardial velocities were recorded using Doppler tissue imaging with the sample volume placed at the lateral mitral annulus in the 4-chamber view.

In all patients, we recorded an ambulatory 24 h Holter ECG (Lifecard CF, Spacelabs Healthcare, Snoqualmie, WA, USA) and manually analyzed and interpreted the data using the Pathfinder SL software (Spacelabs Healthcare, Snoqualmie, WA, USA).In the 24 h ECG we counted the number of premature atrial complexes, defined as a reduction in the RR interval of ≥25% compared to the previous normal RR interval and with normal QRS morphology. We assessed the minimal RR coupling interval as well as the relative reduction in the coupling interval from the previous normal RR interval (coupling interval index (CI-index)).

## 3. Statistical Analysis 

Continuous variables are expressed as means with standard deviations, and categorical variables as numbers with percentages. Continuous variables were compared using the Mann–Whitney U test or t-test in the case of two-group comparison. Differences in proportions were tested with Pearson’s χ^2^ test or Fisher’s exact test, as appropriate. Uni- and multivariable linear regression analysis was conducted to assess the effect of age, sex and group assignment (HCM, HHD and athletes) on the following variables: filtered P-wave duration, PACS count, PR interval, LAVI and BNP. To predict underlying heart disease in patients with HCM and in patients with HHD and an interventricular septum thickness ≥13 mm, a receiver operating characteristic (ROC) analysis was performed to identify cut-off values for variables, which appeared to be different among the groups. The following variables were analyzed: filtered P-wave duration, LAVI, left ventricular end-diastolic diameter and thickness of the interventricular septum. We did not perform this kind of analysis in athletes with an interventricular septum thickness ≥13 mm because of the low number of subjects in this group (*n* = 12). All tests were performed at a two-sided 5% significance level with 95% confidence intervals (CIs). All analyses were performed using Stata 16.1 (StataCorp. Stata Statistical Software: Release 16. College Station, TX, USA: StataCorp LLC).

## 4. Results 

We included 27 HCM patients in the study (30% female; mean age 50 ± 14 years). These patients were compared to 324 patients with HHD (48% female; mean age 75 ± 5.5 years) and to 215 non-elite marathon runners (28% female; mean age 42 ± 7.5 years). Table 1 shows baseline patient characteristics of the three groups. HCM patients were significantly older than athletes and significantly younger than HHD patients (both *p* values < 0.001). The interventricular septum was thicker in HCM patients compared to athletes and HHD patients (Table 1 and Figure 1, *p* < 0.001), and the left ventricular end-diastolic diameter was smaller in HCM patients compared to HHD patients and athletes (Table 1 and Figure 2, *p* < 0.001). In the HHD group 114 patients (35%) and in the athlete group 12 subjects (6%) had an interventricular septum thickness of ≥13 mm (IVS13; Figure 1).

### 4.1. Electrical Markers of Atrial Cardiomyopathy

Filtered P-wave duration was significantly longer in HCM patients compared to HHD patients and athletes (Table 1 and Figure 3). Among athletes with IVS13, filtered P-wave duration was significantly shorter compared to those of HCM patients, and there was a trend for shorter filtered P-wave duration in HHD patients with IVS13 compared to those of HCM patients (Appendix A). Filtered P-wave duration was significantly associated with both older age and male sex (Table 2). In multivariable regression analysis, filtered P-wave duration remained significantly associated with older age and male sex, as well as presence of HCM. Compared to HCM patients, PACs per hour were significantly more frequent in both HHD patients and HHD patients with IVS13 (Table 1 and Appendix A). PACs were rare in athletes. PACs per hour were significantly associated with age and group classification in univariable regression analysis (Table 2). In multivariate regression analysis, only the presence of HHD remained significantly associated with PACs per hour. PR interval was longer in HCM patients compared to those of HHD patients and athletes (Table 1). PR interval was associated with older age and male sex. When corrected for age and sex, the PR interval remained significantly associated with HCM group classification (Table 2).

### 4.2. Echocardiographic Markers of Atrial Cardiomyopathy

Left atrial volume index was larger in HCM patients compared to HHD patients and athletes, and also when compared to HHD patients and athletes with IVS13 (Table 1 and Appendix A). Age and sex were not associated with LAVI, and in multivariate regression analysis LAVI remained associated with HCM group classification (Table 2).

### 4.3. Biochemical Markers of Atrial Cardiomyopathy

Brain natriuretic peptide was higher in HCM patients compared to both HHD patients and HHD patients with IVS13 (Table 1 and Appendix A). High-sensitivity Troponin T and high-sensitivity CRP were not different among HCM and HHD patients. In multivariable regression analysis, brain natriuretic peptide was associated with increasing age and HCM group classification.

### 4.4. Differentiation of HCM Patients from HHD Patients with IVS13

In ROC analysis, the area under the curve to differentiate HHD patients with IVS13 from HCM patients was 0.863 and 0.733, respectively, for the IVS thickness and for LAVI (Table 3 and Figure 4). Using LAVI at a cut-off point of 31 mL/m^2^ yielded a sensitivity and specificity of 81% and 57%, respectively, whereas using IVS thickness at a cut-off point of 14 mm achieved a sensitivity and specificity of 85% and 72%, respectively. Choosing an interventricular septum thickness at a cut-off point of 16 mm resulted in a sensitivity and specificity of 52% and 94%, respectively.

## 5. Discussion

The main findings of our study are as follows: (1) among HCM and HHD patients and athletes, electrical and structural remodeling of the atrium as assessed by filtered P-wave duration, PR interval and LAVI is the most advanced in HCM patients; (2) PACs per hour are most frequent in HHD patients; (3) the prevalence of subjects with IVS13 is 6% in endurance athletes and 35% in an elderly hypertensive patient population; and (4) LAVI ≥31 mL/m^2^ has a sensitivity of 81%, and interventricular septum thickness ≥16 mm a specificity of 94% to differentiate HCM from HHD patients with IVS13.

Compared to the general population, the prevalence of atrial fibrillation is higher in patients with HCM or HHD or in endurance athletes [13,14]. Unfavorable atrial remodeling imposed by chronically increased loading conditions is among the most important risk factors for atrial fibrillation. In HCM, increased ventricular mass and impaired diastolic function is a direct consequence of genetically mediated myocardial disease, and in HHD it is the chronically increased afterload that leads to these conditions. Both pathologies result in increased filling pressure and dilation of the left atria. Additionally, myocyte hypertrophy and disarray probably also directly affect the atrial tissue in HCM [16,17]. In athletes, chronic volume overload with repetitive episodes of atrial stretching during long-term endurance training is mainly responsible for atrial remodeling [18].

Prolonged P-wave duration, as assessed by SAECG, is a correlate for electric atrial remodeling and probably atrial cardiomyopathy and results from a combination of slowed electrical conduction velocity, suggesting the presence of fibrosis, and atrial dilation [27]. In a general population aged 50 years, the filtered P-wave duration is expected to be about 120 ms, [28] and this is clearly shorter than the 153 ms observed in our HCM group. A P-wave duration of 146 ms has been described in patients with HHD and atrial fibrillation, and of 128 ms in HHD patients without atrial fibrillation [29]. Filtered P-wave duration increases with age and is longer in men [30]. After correcting for age and male sex, P-wave duration was still significantly longer in our HCM group compared to the HHD and athlete groups. In fact, after correction for age and male sex, P-wave duration was on average 16 ms longer in HCM group compared to the other two groups. Prolonged PR interval is another well-described predictor of incidental atrial fibrillation and is associated with atrial cardiomyopathy [31]. Determinants of PR interval are conduction time through the right atrium, AV node and His–Purkinje system. Vagal tone profoundly affects PR interval, and is most pronounced in athletes. After correction for age and male sex, PR interval was shortest in the HHD group, and, unsurprisingly, longer in athletes. However, the longest PR intervals were found in the HCM group, with intervals on average 39 ms longer compared to HHD group and 18 ms longer compared to athletes, which could be explained by the structural changes in the myocytes linked to this genetic disease.

Frequent PACs per hour suggests the presence of a trigger for atrial fibrillation [32]. Interestingly, PACs per hour were most numerous in the HHD group, even after adjustment for age and sex. Athletes on the other hand had a low PAC count. An initial sharp increase in the risk of atrial fibrillation with rising PAC count has been reported, starting already at as few as four to five PACs per hour, ref. [32] as observed in our HCM group.

Indexed left atrial volume was largest in the HCM group, and remained unchanged after correction for age and sex. Increased left atrial volume is an established structural criterion for advanced atrial cardiomyopathy, and is also associated with an increased incidence of atrial fibrillation [33]. On average, LAVI was larger by over 12 mL/m^2^ in HCM patients, compared to HHD patients and athletes, in which LAVI was only mildly abnormal. The severely abnormal LAVI in HCM patients reflects high filling pressures due to diastolic dysfunction, and mitral regurgitation following systolic anterior movement of one or both leaflets [34]. To the same end, increased BNP levels in HCM patients also results from abnormal loading conditions, and, after adjustment for age and sex, are increased in HCM compared to HHD patients.

The diagnosis of HCM is based on a wall thickness of ≥15 mm anywhere in the left ventricle in the absence of another cause of hypertrophy. Diagnosis of HCM can be difficult in HCM patients and athletes, particularly in the presence of an interventricular septum with a borderline thickness of 1–14 mm. In our cohort of endurance athletes, such borderline interventricular septum thickness was present in 6% of cases, similar to the rates published in other series [35]. Compared to HCM patients, athletes and HHD patients tend to have a larger left ventricular cavity relative to septum thickness, which was also the case in our study [36]. Mildly dilated atria are a frequent finding in athletes and patients with HHD, whereas atria are more severely dilated in HCM patients [37]. Accordingly, a LAVI ≥31 mL/m^2^ had a sensitivity of 81% for distinguishing HCM patients from patients with HHD and IVS13, whereas an interventricular septum thickness ≥16 mm was highly specific.

The findings of this study should be viewed in light of several limitations. First, there were significant differences in patient characteristics between the three groups studied. Particularly, HCM patients and athletes were more likely to be male and younger than HHD patients. Second, although data was collected prospectively for all three groups by the same investigators and using the same methods, evaluations were performed at varying times, which may have introduced bias. Third, we included patients with an established diagnosis of HCM for this study. These patients tended to have a more advanced phenotype of HCM, and the results may not apply to patients with an inconclusive diagnosis or less advanced phenotype. Finally, the HCM group included a much smaller number of cases as compared to the other two groups. Larger datasets might allow for analysis of more variables affecting atrial remodeling.

## 6. Conclusions

Our findings indicate that electrical and structural remodeling of the atrium is more advanced in HCM patients compared to HHD patients and athletes. These findings might be attributed to the complex pathophysiology of HCM with the involvement of a genetic factor and a remodeling secondary to reduced LV compliance.

## Figures and Tables

**Figure 1 jcm-11-01316-f001:**
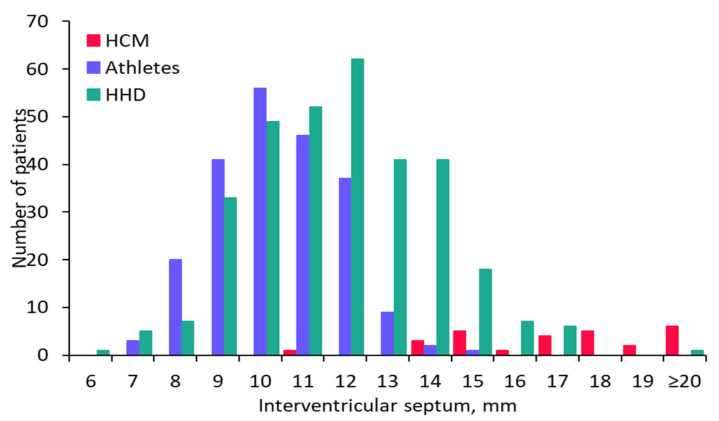
Thickness of the interventricular septum in the three groups. Bar graph showing the thickness of the interventricular septum in millimeters on the x-axis and the corresponding number of patients in each of the three groups on the y-axis. The dashed line separates patients with a thickness of the interventricular septum ≥ 13 mm from the remaining patients. HCM: hypertrophic cardiomyopathy; HHD: hypertensive heart disease.

**Figure 2 jcm-11-01316-f002:**
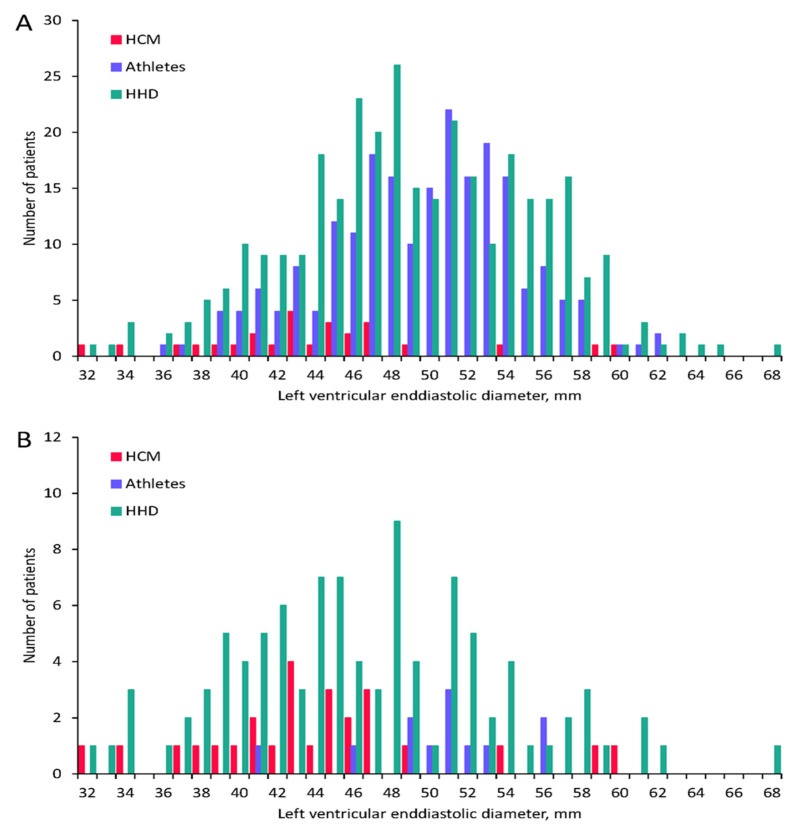
Left ventricular end-diastolic diameter in all patients and in those with septal thickness >13 mm. Bar graph showing the left ventricular end-diastolic diameter in millimeters on the x-axis and the corresponding number of patients in each of the three groups on the y-axis for all patients (**A**) and for patients with an interventricular septum thickness ≥13 mm only in the HHD and athlete groups (**B**). HCM: hypertrophic cardiomyopathy; HHD: hypertensive heart disease.

**Figure 3 jcm-11-01316-f003:**
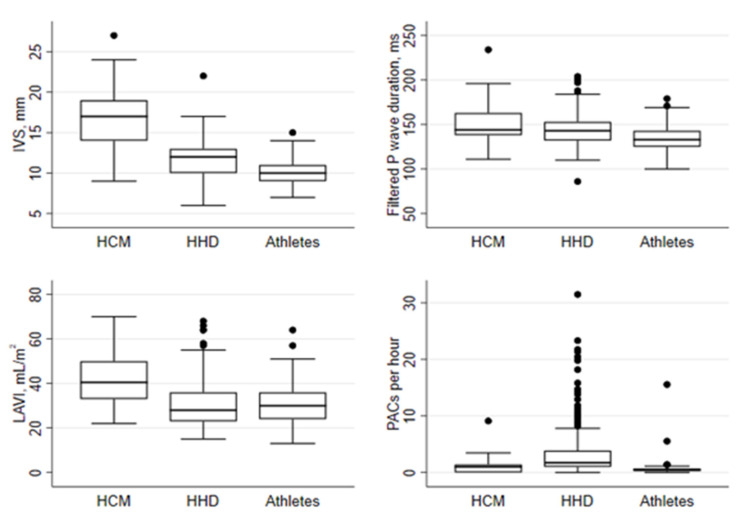
Comparison of the interventricular septal thickness, filtered P-wave duration, LAVI and PACs per hour between the groups. Boxplots for HCM, HHD and athlete groups showing IVS, filtered P-wave duration, LAVI and square-root transformed PACs per hour. IVS: interventricular septum; LAVI: left atrial volume index; PACs: premature atrial complexes.

**Figure 4 jcm-11-01316-f004:**
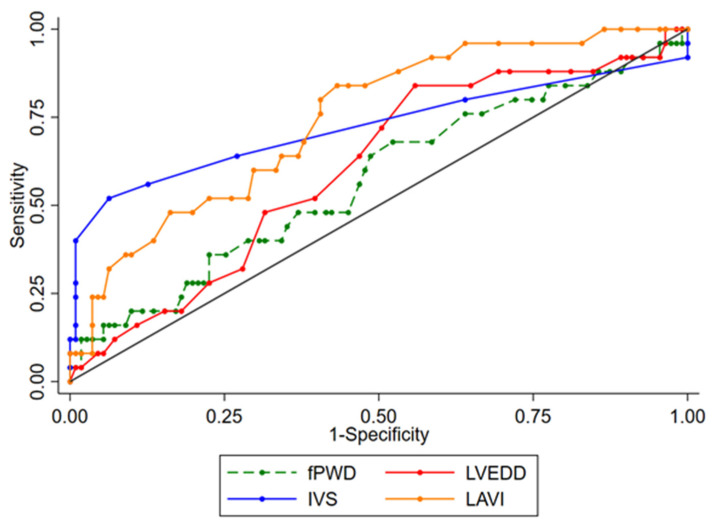
Receiver operating characteristic curves for distinguishing HCM patients from HHD patients with an interventricular septum thickness ≥13 mm. FPWD: filtered P-wave duration; IVS: interventricular septum thickness; LAVI: left atrial volume index; LVEDD: left ventricular diastolic diameter.

**Table 1 jcm-11-01316-t001:** Patient characteristics.

	HCM *n* = 27	HHD *n* = 324	*p* Value *	Athletes *n* = 215	*p* Value **
**Clinical characteristics**					
Age, years	50 ± 14	75 ± 5.5	<0.001	42 ± 7.5	<0.001
Sex, female	8 (30%)	155 (48%)	0.074	61 (28%)	1.000
BMI, kg/m^2^	27 ± 4.8	28 ± 4.4	0.667	22 ± 2.2	<0.001
Arterial hypertension	10 (37%)	324 (100%)	<0.001	-	-
Diabetes mellitus	4 (15%)	85 (26%)	0.012	-	-
Dyslipidemia	11 (41%)	209 (65%)	0.269	-	-
Coronary artery disease	2 (7%)	134 (41%)	<0.001	-	-
Congestive heart failure	-	9 (3%)	1.000	-	-
Previous thrombotic event	2 (7%)	57 (18%)	0.282	-	-
Medication					
Betablocker	14 (52%)	155 (48%)	0.842	-	-
Calcium channel blocker	9 (33%)	99 (31%)	0.831	-	-
ACE inhibitors	5 (19%)	97 (30%)	0.272	-	-
ARB	3 (11%)	133 (41%)	0.002	-	-
Aldactone	1 (4%)	10 (3%)	0.602	-	-
Diuretics	4 (15%)	95 (29%)	0.121	-	-
Statins	10 (37%)	200 (62%)	0.013	-	-
**ECG**					
Heart rate, bpm	63 ± 11	67 ± 11	0.042	55 ± 7.8	<0.001
PR interval, ms	191 ± 48	178 ± 31	0.034	165 ± 26	<0.001
QRS width, ms	110 ± 27	97 ± 20	0.002	95 ± 10	<0.001
QTc, ms	448 ± 27	439 ± 24	0.059	414 ± 23	<0.001
**24 h Holter ECG**					
Minimal heart rate, beats per minute	54 ± 8.4	54 ± 7.6	0.570	45 ± 6.3	<0.001
Number of PACs per hour	4.9 ± 16	27 ± 86	<0.001	2.7 ± 23	0.639
**SAECG**					
Filtered P-wave duration, ms	153 ± 26	144 ± 16	0.012	134 ± 14	<0.001
RMS voltage of P wave, μV	7.5 ± 2.4	6.4 ± 2.3	0.018	7.7 ± 2.6	0.648
P-wave integral, μVs	850 ± 272	672 ± 235	<0.001	773 ± 260	0.153
RMS voltage of terminal 20 ms, μV	3.8 ± 1.4	4.3 ± 2.4	0.320	4.6 ± 2.2	0.073
RMS voltage of terminal 30 ms, μV	3.9 ± 1.3	4.3 ± 2.4	0.332	4.7 ± 2.3	0.073
RMS voltage of terminal 40 ms, μV	5.2 ± 1.8	5.5 ± 8.6	0.878	5.7 ± 2.7	0.330
**Laboratory**					
BNP, pg/mL	142 ± 126	84 ± 93	0.002	-	-
hsTNT, μg/L	0.02 ± 0.02	0.01 ± 0.02	0.391	-	-
hsCRP, mg/L	2.8 ± 4.0	3.1 ± 3.9	0.726	-	-
**Echocardiography**					
LVEF, %	66 ± 7.0	61 ± 6.9	<0.001	65 ± 5.3	0.475
LVEDD, mm	44 ± 6.4	49 ± 6.3	<0.001	50 ± 4.9	<0.001
IVS, mm	18 ± 3.4	12 ± 2.2	<0.001	10 ± 1.5	<0.001
PW, mm	11 ± 3.3	11 ± 1.9	0.318	10 ± 1.3	<0.001
LVMI, g/m^2^	154 ± 56	133 ± 30	0.003	99 ± 20	<0.001
RVD, mm	32 ± 5.7	32 ± 4.5	0.947	36 ± 3.8	<0.001
RV TDI S, cm/s	13 ± 2.8	13 ± 3.1	0.230	14 ± 2.1	0.001
RV diastolic area, cm^2^	18 ± 3.5	17 ± 4.6	0.280	23 ± 3.6	<0.001
LAVI, ml/m^2^	43 ± 14	30 ± 10	<0.001	31 ± 9.5	<0.001
E wave, cm/s	71 ± 20	67 ± 20	0.353	77 ± 13	0.042
A wave, cm/s	62 ± 25	87 ± 20	<0.001	51 ± 12	<0.001
E wave/A wave ratio	1.4 ± 1.0	0.8 ± 0.3	<0.001	1.6 ± 0.4	0.123
Isovolumetric relaxation time, ms	101 ± 13	97 ± 23	0.501	84 ± 13	<0.001
E deceleration time, ms	222 ± 60	259 ± 66	0.006	175 ± 29	<0.001
E wave TDI, cm/s	5.8 ± 1.8	5.7 ± 1.6	0.722	10 ± 1.8	<0.001
E wave/E wave TDI ratio	14 ± 10	12 ± 4.5	0.144	7.5 ± 1.5	<0.001
A wave TDI, cm/s	7.2 ± 2.1	10 ± 2.2	<0.001	8.7 ± 1.7	<0.001

Shown are numbers with percentages in parentheses, or means ± standard deviations, as appropriate. * Comparing HCM versus HHD patients. ** Comparing HCM patients versus athletes. ACE: angiotensin converting enzyme; ARB: angiotensin receptor blocker; BMI: body mass index; BP: blood pressure; CAD: coronary artery disease; CCB: calcium channel blocker; ECG: electrocardiogram; HCM: hypertrophic cardiomyopathy; HHD: hypertensive heart disease; PACS: premature atrial contractions; RMS: root-mean-square; SAECG: signal-averaged ECG; PW: posterior wall; LVMI: left ventricular mass index; RVD: right ventricular diameter.

**Table 2 jcm-11-01316-t002:** Uni- and multivariable linear regression analysis to assess the effect of age, sex and group assignment on fPWD, PACS count, PR interval, LAVI and BNP.

	Age	*p* Value	Sex (Female)	*p* Value	HCM vs. HHD	*p* Value	HCM vs. Athletes	*p*-Value
**Univariable**								
fPWD, ms	0.3 (0.2 to 0.4)	<0.001	−7.7 (−10.6 to −4.7)	<0.001	−8.8 (−18.6 to 1.0)	0.079	−19.2 (−29.0 to −9.3)	<0.001
PACs per hour, sqrt	0.3 (0.3 to 0.4)	<0.001	−3.2 (−6.4 to 0.0)	0.053	8.8 (5.0 to 12.6)	<0.001	−3.9 (−7.3 to −0.5)	0.026
PR interval, ms	0.5 (0.3 to 0.6)	<0.001	−7.6 (−12.6 to −2.5)	0.003	−13.7 (−31.7 to 4.3)	0.135	−26.3 (−44.3 to −8.2)	0.004
LAVI, mL/m^2^	−0.0 (−0.1 to 0.0)	0.219	0.6 (−1.3 to 2.5)	0.537	−12.8 (−18.1 to −7.4)	<0.001	−12.5 (−18.0 to −6.9)	<0.001
BNP, pg/mL	0.3 (−1.3 to 1.8)	0.738	15.4 (−5.1 to 35.8)	0.140	−59 (−107 to −11)	0.017	-	-
**Multivariable**								
fPWD, ms	0.4 (0.2 to 0.6)	<0.001	−9.4 (−12.0 to −6.7)	<0.001	−16.9 (−26.3 to −7.5)	<0.001	−16.3 (−26.2 to −6.3)	0.001
PACs per hour, sqrt	0.0 (−0.2 to 0.3)	0.804	−3.1 (−6.2 to 0.0)	0.053	8.5 (0.8 to 16.1)	0.029	−2.9 (−7.8 to 1.9)	0.238
PR interval, ms	1.1 (0.7 to 1.5)	<0.001	−10.1 (−14.9 to −5.2)	<0.001	−38.5 (−59.2 to −17.7)	<0.001	−18.4 (−33.8 to −3.0)	0.020
LAVI, mL/m^2^	0.1 (−0.1 to 0.2)	0.395	1.0 (−0.8 to 2.9)	0.253	−14.6 (−21.0 to −8.2)	<0.001	−12.3 (−18.0 to −6.5)	<0.001
BNP, pg/mL	2.9 (1.3 to 4.6)	0.001	18.9 (−0.6 to 38.5)	0.058	−135.2 (−190.7 to −79.8)	<0.001	-	-

Shown are average differences/effects with 95% confidence intervals. BNP: brain natriuretic peptide; fPWD: filtered P-wave duration; HCM: hypertrophic cardiomyopathy; HHD: hypertensive heart disease; LAVI: left atrial volume index; PACs: premature atrial complexes; sqrt: square-root transformed.

**Table 3 jcm-11-01316-t003:** ROC curve analyses.

	Variable	AUC (95%-CI)	Cut-Off Point	Youden Index	Sensitivity	Specificity	*p*-Value
HCM vs. HHD	fPWD, ms	0.587 (0.461,0.712)	139	0.177	0.704	0.474	0.176
HCM vs. HHD	LVEDD, mm *	0.610 (0.494,0.727)	48	0.285	0.846	0.439	0.088
HCM vs. HHD	IVS, mm	0.863 (0.773,0.952)	14	0.571	0.852	0.719	<0.001
HCM vs. HHD	LAVI, mL/m^2^	0.733 (0.633,0.832)	31	0.382	0.815	0.568	<0.001

* Indicating HCM, if the value is equal to or lower than the cut-off point, otherwise indicating HHD. fPWD: filtered P-wave duration; IVS: interventricular septum thickness; LAVI: left atrial volume index; LVEDD: left ventricular diastolic diameter.

## Data Availability

Individual patient data included in this study can not be provided due to restrictions set by the Ethics Committee.

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
