# Peer review of "Differences in Atrial Remodeling in Hypertrophic Cardiomyopathy Compared to Hypertensive Heart Disease and Athletes’ Hearts"

_jcm, 2022, doi:10.3390/jcm11051316_

Round 1

Reviewer 1 Report

The article is well written with relevant findings. Results are well presented and sustained by organised figures and tables.

I would suggest adding a  Conclusion paragraph.

Author Response

We thank the reviewer for positive comments. In fact, our paper does include a conclusion statement. We have now expanded the conclusions. It reads as follows:

“Conclusion

Our findings indicate that electrical and structural remodeling of the atrium is more advanced in HCM patients compared to HHD patients and athletes. These findings might be attributed to the complex pathophysiology of HCM with the involvement of a genetic factor and a remodeling secondary to reduced LV compliance.”

Reviewer 2 Report

The article contains an evaluation of left atrial remodeling in three different groups of patients with hypertension, hypertrophic cardiomyopathy and athletes. The work is carefully prepared and I have no objections to its presentation. The research is not novel and confirms known findings about left atrial remodeling. It would be interesting to observe how the presented changes affect the prognosis. Other limitations, that the authors also mention, include the different and small number of groups. It would be interesting to increase the number of groups and follow up observation, which would enrich the value of the article.

Author Response

The reviewer points into the fact that the findings presented in our manuscript are not entirely novel and partly reflect the common knowledge. We agree with this statement. The objective of this paper was not to assess the impact of atrial remodeling on the prognosis of these patients. In fact, for such an analysis the study should have had a completely different design and much larger number of HCM patients. Regarding the small number of HCM patients, we agree this is a limitation and therefore added it as a limitation.

Finally, the HCM group included much smaller number of cases as compared to the other two groups. Larger datasets might allow for analysis of more phenotype modifiers.”

Reviewer 3 Report

The study design is good, it is well described, it is of interest to professionals working in this area, there are limitations that you have already detected, but without compromising the study. I suggest that you seek to solve these possible biases in a future study.

The present study evaluated patients with hypertrophic cardiomyopathy, hypertensive heart disease and athlete's heart by analyzing the prevalence of atrial fibrillation in the three groups. The hypothesis was that there could be different characteristics in these atrial heart diseases.

The methodology was well defined and the results detected significant differences in the groups. The conclusion was precise and objective.

The authors were careful to identify possible bias in their study and made these observations. One of them concerns the period of data collection. Therefore, I suggested to the authors to take these precautions in future studies, as this is already concluded and suggested for publication.

The strengths were the punctual analysis of these groups of patients, the identification of differences suggested in the hypothesis, which brings a contribution to the professionals who work with these patients and the good writing of the study. Weak point is the possibility of bias described by the author himself.

Despite the possibility of these bias, the results were significant and they will hardly interfere with the results and I did not consider them to be important.

Author Response

We thank the reviewer for positive evaluation of our study